# Cancers adapt to their mutational load by buffering protein misfolding stress

Susanne Tilk[1]*, Judith Frydman[1], Christina Curtis[2,3,4]*, Dmitri A Petrov[1,4]*

[1]Department of Biology, Stanford University, Stanford, United States; [2]Department of Medicine, Division of Oncology, Stanford University School of Medicine, Stanford, United States; [3]Department of Genetics, Stanford University School of Medicine, Stanford, United States; [4]Stanford Cancer Institute, Stanford University School of Medicine, Stanford, United States

## eLife assessment

Tilk and colleagues present a computational analysis of tumor transcriptomes to investigate the hypothesis that the large number of somatic mutations in some tumors is detrimental such that these detrimental effects are mitigated by an up-regulation by pathways and mechanisms that prevent protein misfolding. The authors address this question by fitting a model that explains the log expression of a gene as a linear function of the log number of mutations in the tumor and show that specific categories of genes (proteasome, chaperones, ...) tend to be upregulated in tumors with a large number of somatic mutations. Some of the associations presented could arise through confounding, but overall the authors present **solid** evidence that mutational load is associated with higher expression of genes involved in mitigation of protein misfolding – an **important** finding with general implications for our understanding of cancer evolution.

**\*For correspondence:**
tilk@stanford.edu (ST);
cncurtis@stanford.edu (CC);
dpetrov@stanford.edu (DAP)

**Abstract** In asexual populations that don't undergo recombination, such as cancer, deleterious mutations are expected to accrue readily due to genome-wide linkage between mutations. Despite this mutational load of often thousands of deleterious mutations, many tumors thrive. How tumors survive the damaging consequences of this mutational load is not well understood. Here, we investigate the functional consequences of mutational load in 10,295 human tumors by quantifying their phenotypic response through changes in gene expression. Using a generalized linear mixed model (GLMM), we find that high mutational load tumors up-regulate proteostasis machinery related to the mitigation and prevention of protein misfolding. We replicate these expression responses in cancer cell lines and show that the viability in high mutational load cancer cells is strongly dependent on complexes that degrade and refold proteins. This indicates that the upregulation of proteostasis machinery is causally important for high mutational burden tumors and uncovers new therapeutic vulnerabilities.

## Introduction

Cancer develops from an accumulation of somatic mutations over time. While a small subset of these mutations drive tumor progression, the vast majority of remaining mutations, known as passengers, don't help and might hinder cancer growth. The role that passengers play in tumor progression has traditionally received little attention despite their abundance and variation across cancer types. The number of passengers in a tumor can vary by over four orders of magnitude, even within the same cancer type, from just a few to tens of thousands of point mutations (*Lawrence et al., 2013*).

**eLife digest** How and when cells grow and divide is tightly controlled. Over time, the DNA in cells accumulates changes known as mutations, which sometimes enable cells to ignore control mechanisms and grow into a tumor. While some mutations drive cancer development, most do not and are considered 'passenger' mutations. The number of these mutations can vary greatly between different cancers, with some tumors containing only a few while others may have tens of thousands.

There is ongoing debate about whether passenger mutations are neutral or harmful to tumors. For example, mutations in genes that encode proteins may prevent the proteins from folding correctly into their usual three-dimensional shapes, or otherwise disrupt how they work. Therefore, Tilk et al. wanted to understand how tumors survive despite accumulating thousands of passenger mutations.

To do so, Tilk et al. analyzed the activities of genes in over 10,000 human tumors as they accumulated mutations in protein-coding genes. This analysis revealed that genes encoding regulatory proteins that help to manage incorrectly folded proteins were more active in tumors with many mutations in protein-coding genes compared with tumors with fewer mutations in such genes. Further experiments in cancer cells grown in a laboratory demonstrated that these mechanisms that degrade and refold abnormal proteins, are essential for cancer survival.

The findings indicate that, while cancer cells survive by acquiring mutations that promote their growth, they must also contend with harmful mutations that may disrupt how their proteins work. Balancing the production and degradation of proteins is critical for cancer cells to survive. Identifying how to disrupt this balance could be used to develop new cancer treatments in the future.

Whether these passengers are neutral or damaging to tumors has long been a matter of debate (*Bakhoum and Landau, 2017*; *Martincorena et al., 2017*; *Bozic et al., 2019*; *McFarland et al., 2017*; *McFarland et al., 2013*; *Zapata et al., 2018*; *Williams et al., 2016*; *Cannataro and Townsend, 2018*; *Tilk et al., 2022*). Some have argued that passengers are functionally unimportant to tumors given that most non-synonymous mutations are not removed by negative selection in somatic tissues (*Bakhoum and Landau, 2017*; *Martincorena et al., 2017*). This is in direct contrast to the human germ line, where non-synonymous mutations are functionally damaging to most genes (*Eyre-Walker and Keightley, 2007*) and signals of negative selection are pervasive (*Martincorena et al., 2017*). The common explanation for why damaging protein-coding mutations are removed in the human-germline but maintained in somatic tissues is that most genes are only important for multi-cellular function at the organismal level (e.g. during development), but not during somatic growth (*Bakhoum and Landau, 2017*; *Martincorena et al., 2017*).

However, the notion that non-synonymous mutations are only selectively neutral in somatic tissues is surprising given their known functional consequences in the germ-line. Non-synonymous mutations are known to be damaging in the human germ-line due to their effects on protein folding and stability (*Drummond and Wilke, 2008*), which ought to be shared between somatic and germline evolution. An alternative explanation is that non-synonymous mutations are indeed damaging in somatic evolution, but negative selection is too inefficient at removing them due to linkage effects driven by the lack of recombination in somatic cells (*Tilk et al., 2022*). Without recombination to break apart combinations of mutations, selection must act on beneficial drivers and deleterious passengers that arise in the same genome together. This makes it less efficient for selection to individually favor beneficial drivers or remove deleterious passengers (*Hill and Robertson, 2007*). As a result, a substantial number of weakly damaging passengers can accrue in cancer due to inefficient negative selection over time. In support of this model, tumors with very small numbers of passengers – where linkage effects are expected to be negligible – have recently been shown to exhibit signatures of negative selection and weed out damaging non-synonymous mutations (*Tilk et al., 2022*). In contrast, the remaining majority (>95%) of tumors, which contain much larger numbers of linked mutations, display patterns of inefficient negative selection. This provides evidence in favor of the inefficient selection model and implies that most tumors carry a correspondingly large deleterious mutational load.

If individual passengers are in fact substantially damaging in cancer, successful tumors with thousands of linked mutations must find ways to maintain their viability by mitigating this large mutational load. While paths to mitigation are difficult to predict for non-coding mutations, tumors with

mutations in protein-coding genes are expected to minimize the damaging phenotypic effects of protein misfolding stress. Here, we investigate this hypothesis by analyzing tumor tissues with paired mutational and gene expression profiles to assess how the physiological state of cancer cells change as they accumulate protein-coding mutations. Using a GLMM, we leverage variation across 10,295 tumors from 33 cancer types and find that complexes that re-fold proteins (chaperones), degrade proteins (proteasome) and splice mRNA (spliceosome) are up-regulated in high mutation load tumors. We validate these results by showing that similar physiological responses occur in high mutational load cancer cell lines as well. Finally, we establish a causal connection by showing that high mutational load cell lines are particularly sensitive when proteasome and chaperone function is disrupted through downregulation of expression via short-hairpin RNA (shRNA) knock-down or targeted therapies. Collectively, these data indicate that the viability of high mutational load tumors is strongly dependent on the up-regulation of complexes that degrade and refold proteins, revealing a generic vulnerability of cancer that can potentially be therapeutically exploited.

## Results

### Quantifying transcriptional response to mutational load in human tumors

We first performed a genome-wide screen to systematically identify which genes are transcriptionally upregulated in response to mutational load in human tumors. To do so, we utilized publicly available whole-exome and gene expression data from 10,295 human tumors across 33 cancer types from The Cancer Genome Atlas (TCGA) (*Weinstein et al., 2013*; *Ellrott et al., 2018*). We considered multiple classes of mutations to define mutational load and investigated their degree of collinearity, focusing on protein-coding regions since the use of whole-exome data limits the ability to accurately assess mutations in non-coding regions. We find that there is a high degree of collinearity among synonymous, non-synonymous, and nonsense point mutations in protein-coding genes ($R>0.9$) but weak collinearity between point mutations and copy number alterations ($R<0.05$) (*Figure 1—figure supplement 1*). Thus, we decided to focus on the aggregate effects of protein-coding mutations and for all analyses defined mutational load as $\log_{10}$ of the total number of point mutations in protein-coding genes. For simplicity, we used all mutations rather than focusing only on passenger mutations since identifying genuine drivers against a background of linked passenger events can be difficult, especially for tumors with many mutations.

Since gene expression can vary across tumors due to many factors, such as cancer type, tumor purity, and other unknown factors, we utilized a GLMM to measure the association of mutational load and gene expression while accounting for these potential confounders (*Figure 1A*). Within the GLMM, tumor purity and mutational load were modeled as fixed effects whereas cancer type was modeled as a random effect since it varies across groups of patients and can be interpreted as repeated measurements across groups. The following GLMM was applied separately to each gene,

$$Y \sim \beta_0 + \beta_1 X_1 + \beta_2 X_2 + v + e$$

where $Y$ is a vector of normalized expression values across all tumors, $\beta_0$ is the fixed intercept, $\beta_1$ is the fixed slope for the predictor variable $X_1$ which is a vector of mutational load values for each tumor, $\beta_2$ is the fixed slope for the predictor variable $X_2$ which is a vector of the purity of each tumor, v is the random intercept for each cancer type, and $e$ is a Gaussian error term (Methods).

Using this approach, we applied the GLMM to all tumors in TCGA and identified 5330 genes that are significantly up-regulated in response to mutational load ($\beta_1>0$, FDR<0.05). Next, we linked these genes to cellular function by performing gene set enrichment to known protein complexes CORUM database (*Giurgiu et al., 2019*, *Figure 1B*) and pathways KEGG database (*Tanabe and Kanehisa, 2012*, *Figure 1C*) using gprofiler2 (*Kolberg et al., 2020*). As expected for tumors with many mutations, pathways, and protein complexes related to the cell cycle, DNA replication, and DNA repair were enriched in tumors with a high mutational load. However, some of the most significant enrichment terms were for protein complexes and pathways that regulate translation (mitochondrial ribosomes), protein degradation (proteasome complex), and protein folding (CCT complex/HSP60), consistent with the hypothesis that high mutational load tumors experience protein misfolding stress. Surprisingly, we also found that the spliceosome, a large protein complex that regulates alternative splicing in cells, is

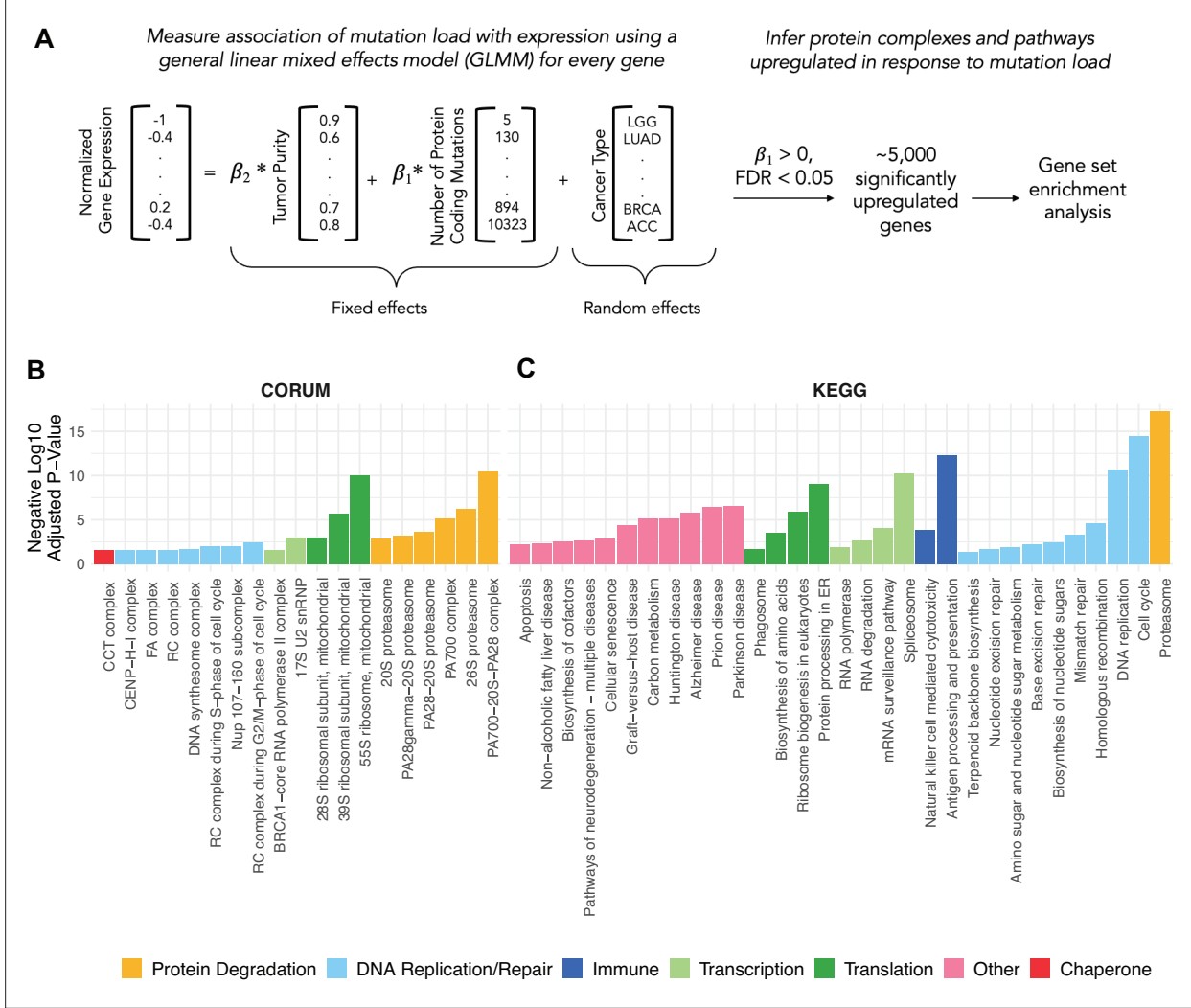

**Figure 1.** General linear mixed effects model (GLMM) identifies protein complexes and pathways up-regulated in response to mutational load in human tumors. (**A**) Overview of the GLMM used to measure the association of mutation load with gene expression while controlling for potential co-variates (purity and cancer type). Genes with a significant, positive $\beta_1$ regression coefficient and false discovery rate (FDR)<0.05 are used for gene set enrichment analysis. (**B**, **C**) Bar plots of protein complexes from the CORUM database (left) and pathways from the KEGG database (right) that are significantly enriched (p<0.05) in response to mutational load. Length of bars denotes negative $\log_{10}$ of adjusted *p*-value and colors denote broad functional groups enriched in both databases.

The online version of this article includes the following figure supplement(s) for figure 1:

**Figure supplement 1.** No collinearity of point mutations and copy number alterations in human tumors (TCGA) and cancer cell lines (CCLE).

**Figure supplement 2.** Genes significantly expressed from the transcriptional screen mostly fall into the upper quartile of effect sizes, which are enriched for proteostasis complexes.

up-regulated in response to mutational load. This suggests that transcription itself could also be regulated in response to protein misfolding stress, as seen in other studies (*Biamonti and Caceres, 2009*; *Dutertre et al., 2011*). In addition, we confirmed that the same proteostasis complexes are identified when performing gene set enrichment analysis only genes with the largest effect sizes from the transcriptional screen (in the upper quartile of $\beta_1$ regression coefficients), which contain half the number of significant genes as identified previously (N=2152 vs 5330; *Figure 1—figure supplement 2*).

## Gene silencing through alternative splicing in high mutational load tumors

We next investigated in detail how these protein complexes could mitigate the damaging effects of protein misfolding in high mutational load tumors by examining the role of the spliceosome in gene silencing. We hypothesized that the up-regulation of the spliceosome in high mutational load tumors prevents further protein misfolding by regulating pre-mRNA transcripts to be degraded rather than translated. The down-regulation of gene expression via alternative splicing events, such as intron retention, is one known mechanism to silence genes by funneling transcripts to mRNA decay pathways (*Ge and Porse, 2014*; *Lindeboom et al., 2016*; *Lareau et al., 2007*).

To test whether gene expression is down-regulated in high mutational load tumors through intron retention, we utilized previously called alternative splicing events in TCGA (*Ryan et al., 2016*). Alternative splicing events within this dataset were quantified through a metric called *percent spliced in* or PSI. PSI is calculated as the number of reads that overlap the alternative splicing event (e.g. for intron retention, either at intronic regions or those at the boundary of exon to intron junctions) divided by the total number of reads that support and don't support the alternative splicing event. Thus, PSI estimates the probability of alternative splicing events only at specific exonic boundaries in the entire transcript population without requiring information on the complete underlying composition of each full-length-transcript.

Using these alternative splicing calls, we reasoned that if a transcript contains an intron retention event and is downregulated in expression, the transcript is more likely to have been degraded by mRNA decay pathways. For all genes, we first quantified whether intron retention events were present based on a threshold value >80% PSI. For each gene with an intron retention event, we quantified whether the expression of the same gene was under-expressed. Each gene was counted as underexpressed if it was one standard deviation below the mean expression within the same cancer type. To control for mutations that might affect patterns of expression, (i.e. expression quantitative trait loci or eQTL effects), alternative splicing events that contained a point mutation within the same gene were removed from the analysis (which only represent ~1% of intron retention events across all

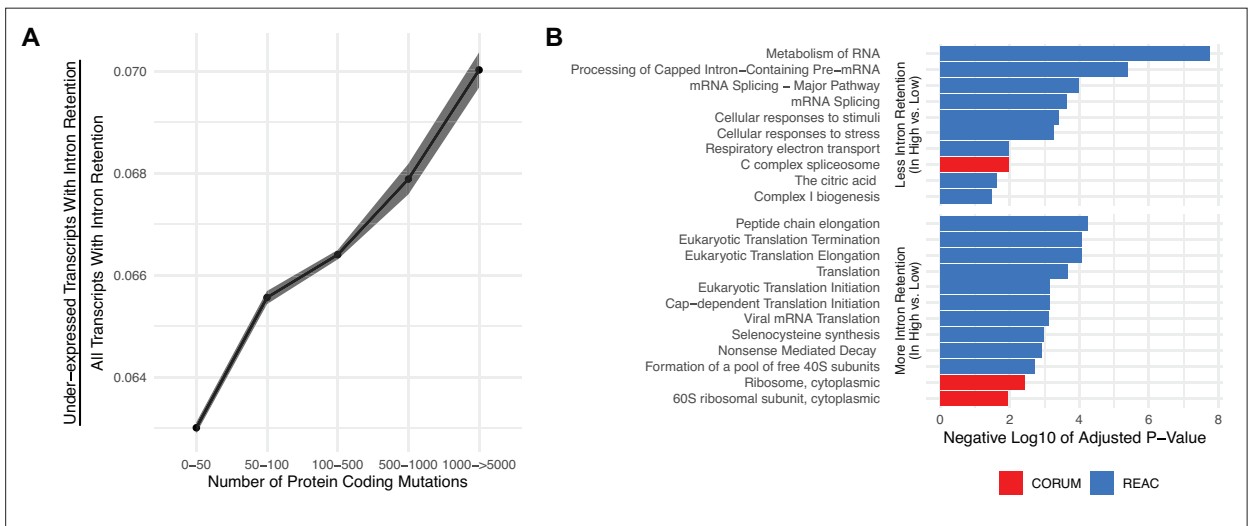

**Figure 2.** Gene silencing is elevated in high mutational load tumors likely through the coupling of intron retention with mRNA decay. (**A**) Counts of the number of under-expressed transcripts with intron retention events, relative to counts of all intron retention events in tumors binned by the total number of protein-coding mutations. Intron retention events with PSI>80% are counted. Error bars are 95% confidence intervals determined by bootstrap sampling. (**B**) Barplot of significant protein complexes in the CORUM database (in red) and Reactome pathway database (in blue) with more (bottom) and less (top) intron retention events in high mutational load tumors compared to low mutational load tumors.

The online version of this article includes the following figure supplement(s) for figure 2:

**Figure supplement 1.** Intron retention events that overlap with mutations do not account for the association of gene silencing in high mutational load tumors.

**Figure supplement 2.** The number of under-expressed transcripts increases with the mutational load of tumors for different PSI value thresholds and alternative splicing events.

tumors; Methods). We find that relative to all transcripts with intron retention events, the number of transcripts that are under-expressed increases with tumor mutational load (*Figure 2A*), suggesting that the degree of intron-retention-driven mRNA decay is elevated in high mutational load tumors. This trend is robust to other PSI value thresholds (>50–90% PSI), even for other alternative splicing events (e.g. exon skipping, mutually exclusive exons, etc.) and when not filtering for potential eQTL effects (*Figure 2—figure supplements 1 and 2*).

We next investigated which genes are more likely to be silenced through mRNA decay between low and high mutational load tumors. For each intron retention event, we calculated whether PSI values were significantly different in low mutational load tumors (<10 total protein-coding mutations) compared to high mutational load tumors (>1000 total protein-coding mutations) using a t-test. This approach identified 606 and 201 genes that have more and less intron retention events in high mutational load tumors, respectively. Using gene set enrichment analysis, we find that cytoplasmic ribosomes contain more intron retention events in high mutational load tumors, potentially leading to their down-regulation through mRNA decay to prevent further protein mis-folding (*Figure 2B*). Genes that contain fewer intron retention events in high mutational load tumors, which are less likely to undergo mRNA decay, are primarily related to mRNA splicing.

## Regulation of translation, protein folding, and protein degradation in high mutational load tumors

Next, we investigated in detail how the remaining proteostasis complexes that were significant in our genome-wide screen, which regulate protein synthesis, degradation, and folding, could mitigate protein misfolding in high mutational load tumors. To do so, we expanded our gene sets to include other chaperone families, all ribosomal complexes, and proteasomal subunits (*Figure 3A*). Using the GLMM framework detailed above, we find that the expression of nearly all individual genes in chaperone families that participate in protein folding (HSP60, HSP70, and HSP90), protein disaggregation (HSP100), and have organelle-specific roles (ER and mitochondrial) are significantly up-regulated in response to mutational load. Interestingly, however, small heat shock proteins, which don't participate in protein folding or disaggregation, are significantly down-regulated in response to increased protein-coding mutations. The role of small heat shock proteins is primarily to hold unfolded proteins in a reversible state for re-folding or degradation by other chaperones (*Sun and MacRae, 2005*) and thus, could possibly be down-regulated due to their inefficiency in mitigating protein misfolding.

We further examined differences in the expression of different structural components of the proteasome, a large protein complex responsible for the degradation of intracellular proteins. Consistent with the over-expression of chaperone families that mitigate protein misfolding, both the 19 s regulatory particle (which recognizes and imports proteins for degradation) and the 20 s core (which cleaves peptides) of the proteasome are up-regulated in response to mutational load in TCGA (*Figure 3A*). In addition, we find that specifically mitochondrial — but not cytoplasmic — ribosome complexes are up-regulated in high mutational load tumors. As previously reported in yeast (*Ruan et al., 2017*) and human cells (*Shcherbakov et al., 2019*), mitochondrial ribosome biogenesis has been shown to occur under conditions of chronic protein misfolding as a mechanism of compartmentalization and degradation of proteins. In contrast, the translation of proteins through cytosolic ribosome biogenesis has been previously characterized to be attenuated and slowed to prevent further protein mis-folding (*Stein and Frydman, 2019*). This decrease in expression of cytoplasmic ribosomes is also consistent with observed patterns of alternative splicing coupled to mRNA decay pathways in high mutational load tumors (*Figure 2B*).

Finally, we performed a jackknife re-sampling procedure to confirm that specific cancer types aren't driving patterns of association within the GLMM. This was achieved by removing each cancer type from the regression model one at a time, and re-calculating regression coefficients on the remaining set of samples. Overall, regression coefficients were stable across cancer types and trends were unchanged (*Figure 3—figure supplement 1*). In addition, we also performed linear regression within cancer types and found similar expression responses to mutational load across proteostasis complexes (*Figure 3—figure supplement 2*). Finally, we also confirmed that patient age was not driving patterns of association of mutational load and gene expression within the GLMM (*Figure 3—figure supplement 3*). Taken together, this suggests that protein re-folding, protein disaggregation, protein degradation,

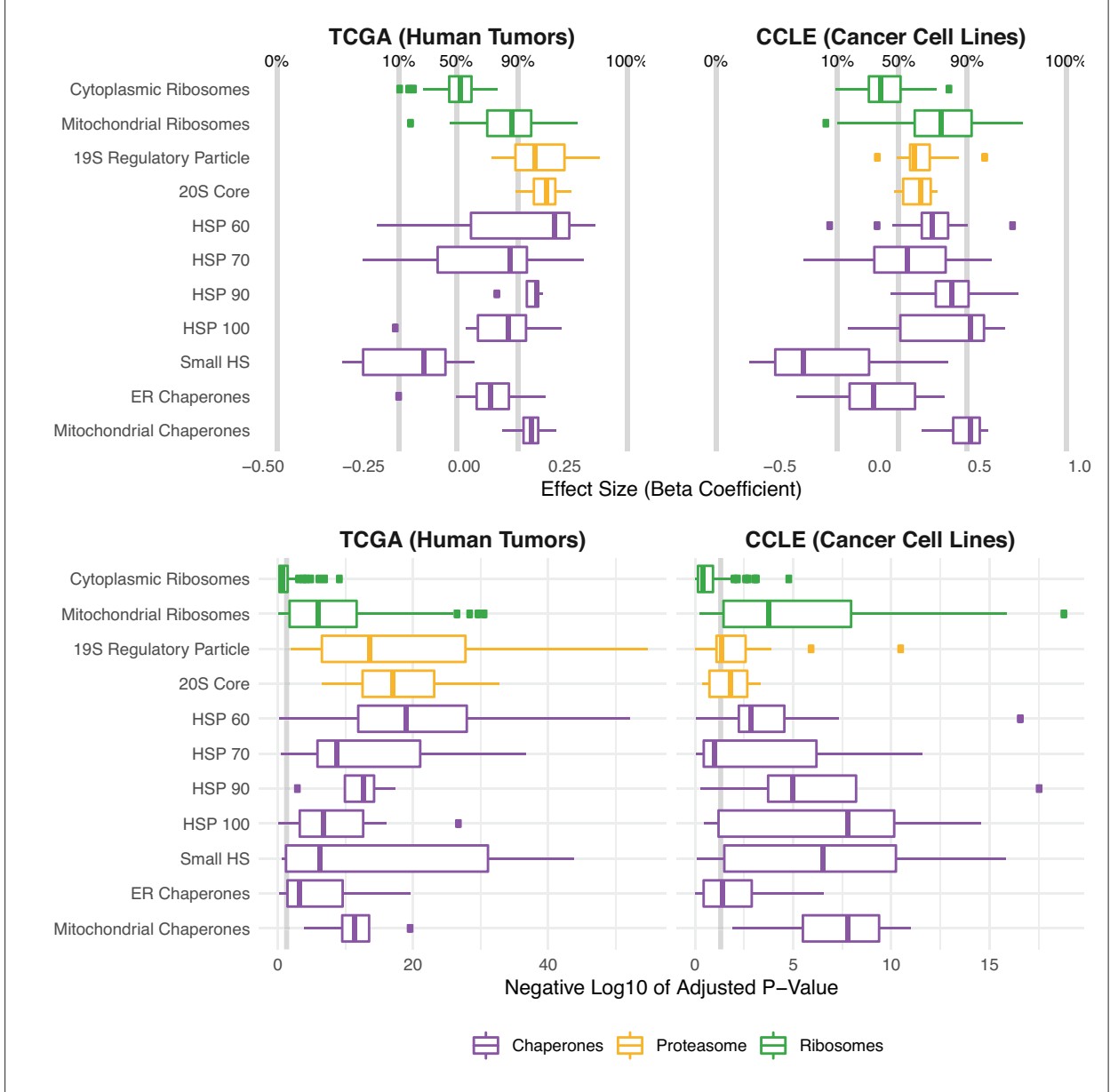

**Figure 3.** Protein folding, degradation, and synthesis are regulated in both high mutational load tumors (TCGA) and cell lines (CCLE). Box plots of $\beta_1$ regression coefficients (top panels) and negative $\log_{10}$ adjusted *p*-values (bottom panels) measuring the association of mutation load and the expression of individual genes in chaperone (purple), proteasome (yellow), and ribosome (green) complexes. Shown are regression coefficients from human tumors (TCGA) on the left and cell lines (CCLE) on the right. Percentages and gray lines on top panels show the quantile distribution of regression coefficients measuring the association of mutational load and expression for all genes in the genome within each dataset. Vertical gray line on the bottom panels shows the threshold of significance (p=0.05).

The online version of this article includes the following figure supplement(s) for figure 3:

**Figure supplement 1.** Association between expression in proteostasis complexes and mutational load is not driven by a single cancer type in The Cancer Genome Atlas (TCGA).

**Figure supplement 2.** Linear regression analysis within cancer types in The Cancer Genome Atlas (TCGA) captures similar expression responses to mutational load across proteostasis complexes.

**Figure supplement 3.** Association between the expression in proteostasis complexes and mutational load is not driven by patient age.

**Figure supplement 4.** Association between the expression in proteostasis complexes and mutational load is not driven by a single cancer type in cancer cell lines (CCLE).

**Figure supplement 5.** Similar patterns of expression and protein abundances in response to mutational load in cancer cell lines (CCLE) within genes that regulate protein folding, degradation, and synthesis.

and down-regulation of cytoplasmic translation are potential mechanisms to mitigate and prevent protein misfolding in high mutational load tumors.

## Validating proteostasis expression responses in cancer cell lines and establishing a causal connection through perturbation experiments

We next sought to validate these results by first examining whether the expression patterns observed in human tumors replicate within cancer cell lines from the Cancer Cell Line Encyclopedia (CCLE) (*Barretina et al., 2012*). Unlike TCGA, samples within each cancer type in CCLE can be small and are unbalanced (i.e. some cancer types have <10 samples and others have >100 samples). Since GLMMs may not be able to estimate among-population variance accurately in these cases (*Harrison et al., 2018*), we utilized a simple generalized linear model (GLM) instead to measure the effect of mutational load on patterns of expression without over-constraining the model. Indeed, we find that expression patterns seen in human tumors broadly replicate in cancer cell lines (*Figure 3*). Similar to the expression analysis in TCGA, we also confirmed through a jackknife re-sampling procedure that specific cancer types aren't driving patterns of association within the GLM (*Figure 3—figure supplement 4*). Finally, we further validated these trends by incorporating protein abundance estimates in CCLE, which contains the largest dataset available of RNA (n=1377) and protein (n=373) abundances that are harmonized across samples. We find similar patterns of expression and protein abundances in response to mutational load in CCLE within proteostasis complexes (*Figure 3—figure supplement 5*).

Overall, this indicated that the expression patterns observed are cell autonomous (i.e. independent of organismal effects such as the immune system, age or microenvironment) and consistent across high mutational load cancer cells. Importantly, it also demonstrates that cancer cell lines are a reasonable model to causally interrogate these effects further through functional and pharmacological perturbation experiments.

To establish a causal relationship between the over-expression of proteostasis machinery and maintenance of cell viability under high mutational load, we utilized expression knock-down (shRNA) estimates from project Achilles (*Tsherniak et al., 2017*) for the same cancer cell lines as in CCLE. We sought to measure how mutational load impacts cell viability when protein complexes and gene families undergo a loss of function through expression knock-down. Since the shRNA screen was performed on an individual gene basis, we utilized a GLM framework that aggregates expression knock-down estimates of all genes within a given proteostasis gene family to jointly measure how mutational load impacts cell viability after loss of function. Specifically, we included an additional categorical variable of the gene name within each gene family to allow for a change in the intercept within each gene in the GLM when measuring the association of mutational load and cell viability after expression knock-down. In addition, we similarly evaluated whether specific cancer types were driving patterns of association within the GLM through jackknife re-sampling by cancer type (*Figure 4A*).

Overall, we find that elevated mutational load is associated with decreased cell viability when the function of most chaperone gene families are disrupted through expression knock-down (*Figure 4A*). However, only chaperones within the HSP100 family, which have the unique ability to rescue and reactivate existing protein aggregates in cooperation with other chaperone families (*Zolkiewski et al., 2012*), show a significant negative relationship between mutational load and cell viability across almost all cancer types. Similarly, we find specificity in the vulnerability that mutational load generates when the function of the proteasome and different ribosomal complexes are disrupted (*Figure 4A*). Mutational load significantly decreases cell viability only when expression knock-down of the 19 s regulatory particle of the proteasome is disrupted, suggesting that targeting the protein import machinery of the proteasome is more effective than targeting the protein cleaving machinery in the 20 s core. Finally, mutational load significantly increases cell viability when cytoplasmic ribosomes – which are already down-regulated in response to mutational load (*Figure 2B*) – undergo a loss of function through expression knock-down. Conversely, expression knock-down of mitochondrial ribosomes significantly decreases viability with increased mutational load in cell lines, which is also consistent with the patterns of expression observed.

Since functional redundancy in the human genome can make expression knock-down estimates within individual genes noisy, we also examined how drugs targeting the function of whole complexes impact viability with mutational load across all cancer types and when removing individual cancer types through jackknife re-sampling. To do so, we utilized drug sensitivity screening data in project

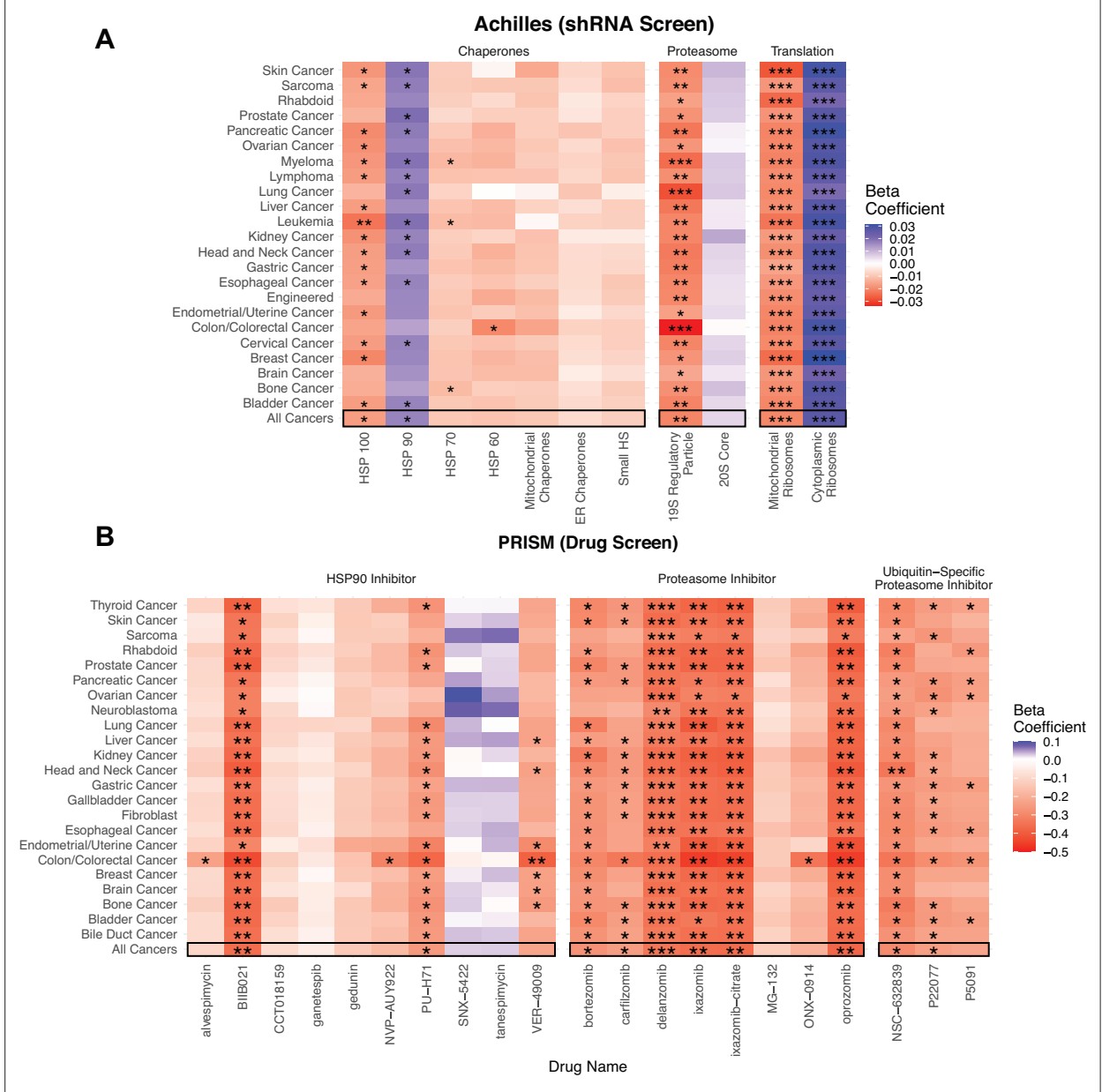

**Figure 4.** Viability in high mutational load cell lines decreases when proteostasis machinery is disrupted. (**A**) Heatmap of $\beta_1$ regression coefficients jointly measuring the association of mutational load and cell viability after expression knockdown of individual genes in proteostasis complexes. (**B**) Heatmap of $\beta_1$ regression coefficients measuring the association mutational load and cell viability after inhibition of proteostasis machinery via drugs. Both panels show how stable regression estimates are when including all cancer types ('All Cancers') shown in black boxes and when removing each individual cancer type on the y-axis. Colors denote a positive (blue), zero (gray), or negative (red) relationship between mutational load and cell viability after expression knockdown or drug inhibition. Stars denote whether the relationship is significant (*p<0.05; **p<0.005; ***p<0.0005).

PRISM (***Corsello et al., 2020***) within CCLE and used a simple GLM to measure the association of mutational load and cell viability after drug inhibition. We find that treatment with the majority of proteasome inhibitors (6/8) and ubiquitin-specific proteasome inhibitors (2/3), which target protein degradation complexes, are significantly associated with a decrease in cell viability in high mutational load cell lines. Similarly, most HSP90 inhibitors decrease cell viability with mutational load (8/10), although only a few drugs show a significant relationship. This variability in the efficacy of drugs with similar mechanisms of action likely reflects that the efficacy to disrupt the function of proteostasis machinery is dependent on the specific molecular affinity of a compound to its target and downstream effectors. While these are the only relevant proteostasis drugs in the PRISM dataset that are currently

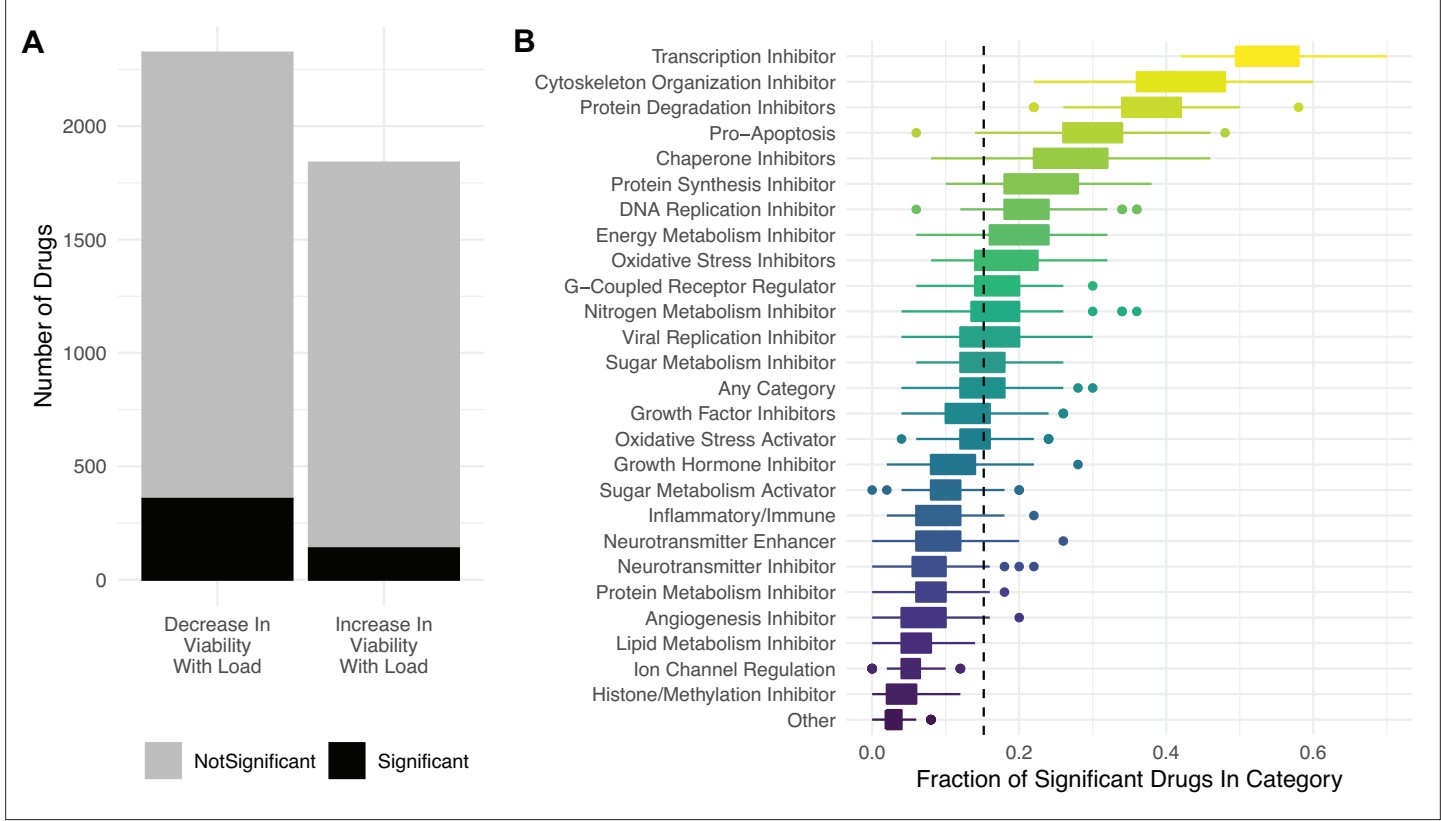

**Figure 5.** Targeting proteostasis machinery is a key vulnerability in high mutational load cell lines. (**A**) Bar plot of the number of drugs in the PRISM database significantly (black) and not significantly (gray) associated with mutational load and cell viability using a simple generalized linear model (GLM). (**B**) Fraction of drugs in broad functional categories significantly negatively associated with mutational load and cell viability from the GLM. Confidence intervals were determined by randomly sampling 50 drugs in each functional category 100 times. Dashed line is the median of randomly sampled drugs across all categories.

available, we anticipate that drugs targeting other chaperone machinery or splicing complexes could also target other potential vulnerabilities in high mutational load cancers. Collectively, these results indicate that elevated expression of protein degradation and folding machinery is causally related to the maintenance of viability in high mutational load cell lines, and likely in high mutational load tumors by extension.

Lastly, we find that most drugs in the PRISM database do not significantly decrease cell viability with mutational load (**Figure 5A**), suggesting that high mutational load cancer cells are not generically vulnerable to all classes of drugs. Specifically, we find that drugs which inhibit transcription, cytoskeleton organization, protein degradation, chaperones, protein synthesis, and promote apoptosis are most effective at targeting high mutational load cancer cells – delineating additional potential therapeutic vulnerabilities in high mutational burden tumors (**Figure 5B**).

## Discussion

Here, we test the hypothesis that cancer cells regulate their proteostasis machinery to mitigate the damaging effects of passenger mutations, which can destabilize and misfold proteins. Misfolded proteins can arise from non-synonymous or nonsense passengers which cause abnormal amino acid modifications or pre-mature truncations in proteins. Even synonymous passengers, which are traditionally thought to be functionally silent, can lead to misfolding of proteins through changes in mRNA stability (**Kristofich et al., 2018**), translational pausing (**Spencer et al., 2012**; **Zhang et al., 2009**), and non-optimal codon usage (**Walsh et al., 2020**; **Plotkin and Kudla, 2011**). As a result, protein misfolding can be damaging in cells not only due to a loss of function of the original protein, but also due to a gain in toxicity caused by the aggregation of aberrant peptides. It is intriguing to consider

the possibility that the need to manage protein misfolding stress is a hallmark of somatic evolution in cancer.

To maintain viability by minimizing these cytotoxic effects, we find that high mutational load tumors – similar to yeast (**Bobula et al., 2006**), bacteria (**Maisnier-Patin et al., 2005**; **Fares et al., 2002**), and viruses (**Elena et al., 2006**) – up-regulate the expression of chaperones, which allow mutated proteins that would otherwise be misfolded to retain function. We find evidence suggesting that specific chaperone families that actively participate in protein re-folding (HSP60, HSP90, and HSP70) or disaggregation (HSP100) are up-regulated in response to mutational load, while other chaperone machinery that salvage proteins (Small HS) are downregulated. In addition, we find degradation of mutated proteins through up-regulation of the proteasome to be another possible strategy high mutational load tumors use to mitigate protein misfolding stress.

Finally, we find additional mechanisms that high mutational load tumors use to not just mitigate but also prevent protein misfolding. By utilizing post-transcriptional processes that couple alternative splicing with mRNA decay pathways known to occur in normal human tissues (**Ge and Porse, 2014**; **Pan et al., 2006**; **Green et al., 2003**), high mutational load tumors appear to selectively prevent protein production by regulating certain pre-mRNA transcripts to be degraded rather than translated. We find evidence suggesting that the targets of this coordinated un-productive splicing are primarily related to cytoplasmic ribosomal gene expression that controls the translation of proteins, consistent with observations in other organisms (**Cuccurese et al., 2005**; **Mitrovich and Anderson, 2000**; **Parenteau et al., 2011**). Intriguingly, we find that while cytoplasmic ribosome expression is attenuated, mitochondrial ribosome biogenesis in human tumors is up-regulated in response to mutational load. This could both be another mechanism that high mutational load tumors use to compartmentalize and degrade proteins (**Ruan et al., 2017**) and reflect the increased energetic demands of proteostasis maintenance (**Kalapis et al., 2015**).

The expression responses observed here are not only consistent with protein misfolding stress in other organisms, but also cross-validate in cancer cell lines, where we find similar expression responses to mutational load. This provides further evidence of a generic, cell-intrinsic phenomenon occurring that cannot be explained by extrinsic organismal effects, such as aging, changes in the immune system, or microenvironment. Furthermore, we move beyond correlations of gene expression responses to mutational load and establish a causal connection by demonstrating that mitigation of protein misfolding through protein degradation and re-folding is necessary for high mutational load cancer cells to maintain viability through perturbation experiments via knockdown experiments with shRNA and drug profiling.

The results presented here have many implications. First, they suggest that while there is direct selection during somatic evolution for pathogenic drivers that allow cancer cells to continually proliferate, damaging passengers that destabilize proteins must also cause cancer cells to experience second-order indirect selection for alterations that allow tumors to overcome this proteostasis imbalance. This could occur through phenotypic plasticity, shifts in methylation and chromatin structure, or through compensatory point mutations and duplications, consistent with other studies (**Tokheim et al., 2021**; **Martínez-Jiménez et al., 2020**). Indeed, gene duplication, where one copy can still perform the required function while the other copy is non-functional, is another known mechanism that allows cells to maintain robustness to damaging mutations in many eukaryotic organisms (**Conant and Wagner, 2004**; **Gu et al., 2003**). In support of this, whole genome duplication, which is common in cancer, has recently been shown as another potential mechanism that tumor cells could use to maintain robustness to deleterious passengers (**López et al., 2020**). However, duplication events are also known to be deleterious due to gene dosage effects that cause protein imbalance (**Torres et al., 2007**), which could further exacerbate protein misfolding. Further experimental studies are needed to distinguish how cancer cells compensate for protein misfolding and the role that genome duplication may play in this process.

Second, the extra demands of proteostasis maintenance present important vulnerabilities in high mutational load cancers that could be exploited. The clinical use of chaperone inhibitors for cancer treatment has been explored for over two decades (**Neckers and Workman, 2012**; **Pacey et al., 2006**; **Kim et al., 2009**) but no study, to our knowledge, has compared the efficacy of chaperone inhibitor use in tumors stratified by mutational load. Similarly, the clinical use of proteasome inhibitors, which are currently only approved for the treatment of multiple myeloma and mantle-cell lymphoma

(*Manasanch and Orlowski, 2017*; *Park et al., 2018*), has not been directed specifically to high mutational load tumors. While the efficacy of proteasome inhibitors in multiple myeloma patients is linked to the protein misfolding stress response (*Bianchi et al., 2009*; *Ling et al., 2012*), it is currently unknown whether high mutational load tumors are more susceptible to these inhibitors. Outside of drugs in the clinic, the need for cancers to compensate for protein misfolding could also present additional vulnerabilities due to evolutionary trade-offs, where the improvement in fitness of one trait comes at the expense of another. Previous work in yeast has identified strong trade-offs between the adaptive mechanisms that allow for the tolerance of mistranslation and survival under conditions of starvation (*Kalapis et al., 2015*). Whether similar conditions could be exploited in high mutational load cancer cells warrants additional further investigation.

Finally, our results contribute to an accumulating body of evidence that cancer and aging are different manifestations of related underlying evolutionary processes (*Martincorena et al., 2018*; *Martincorena et al., 2015*; *Kennedy et al., 2019*). The same forces of mutation and inefficient selection in somatic evolution generate a persistent problem of deleterious mutation accumulation in normal somatic tissues and during tumor growth. Disruption of proteostasis is a known hallmark of aging in normal tissues (*López-Otín et al., 2013*). Many transcriptional responses observed in high mutational load tumors — such as shifts in regulation of alternative splicing (*Adusumalli et al., 2019*), protein degradation (*Löw, 2011*), and protein re-folding (*Soti and Csermely, 2003*) — are also observed in normal aging tissues which contain somatic mutations. Despite this, aging tissues appear to utilize different strategies to deal with proteostasis disruption — such as up-regulation of chaperones in the Small HS family (*Charmpilas et al., 2017*) and autophagy (*Aman et al., 2021*) — which are not a predominant response observed here in high mutational load tumors. Whether different combinations of strategies are used by high mutational load cancer cells use to overcome their mutational load or whether all the strategies identified here are needed to maintain proteostasis is unclear. Differences in these proteostasis strategies could be due to different selection pressures during somatic evolution, the degree of mutational load required to induce a stress response, differences in energetic costs of protein maintenance, or the interplay that exists between apoptosis and proteostasis. Further studies are needed to elucidate the precise dynamics and physiological consequences of inefficient negative selection in somatic evolution, how this impacts cellular growth, and the mechanisms somatic cells use to maintain robustness to proteostasis disruption.

## Methods
### Statistical analysis

The lmerTest and lmer package in R was used to apply a separate generalized linear mixed model (GLMM) for each gene in the genome to identify groups of genes whose expression is up-regulated in response to mutational load in TCGA. For each gene, expression values across all patients were z-score normalized in all analyses to ensure fair comparisons across genes. Known co-variates of tumor purity and cancer type were included in the GLMM. Tumor purity and mutational load were modeled as fixed effects, whereas cancer type was modeled as a random effect (i.e. random intercept) since it varies across groups of patients and can be interpreted as repeated measurements across groups. For all analyses, mutational load was defined as $\log_{10}$ of the number of synonymous, nonsynonymous, and nonsense mutations per tumor. For each gene, the parameters used in the GLMM were as follows,

$$Y \sim \beta_0 + \beta_1 X_1 + \beta_2 X_2 + v + e$$

where $Y$ is a vector of expression values of each tumor, $\beta_0$ is the fixed intercept, $\beta_1$ is the fixed slope for the predictor variable $X_1$ which is a vector of mutational load values for each tumor, $\beta_2$ is the fixed slope for the predictor variable $X_2$ which is a vector of the purity of each tumor, v is the random intercept for each cancer type, and $e$ is a Gaussian error term. To examine expression responses to mutational load within a given protein complex and cancer type, the same normalization procedures were applied as above within cancer types and a separate GLM for each cancer type was ran as follows,

$$Y \sim \beta_0 + \beta_1 X_1 + \beta_2 X_2 + \beta_3 X_3 + e$$

where $Y$ is a vector of expression values of each tumor in a given cancer type, $\beta_0$ is the fixed intercept, $\beta_1$ is the fixed slope for the predictor variable $X_1$ which is a vector of mutational load values for each tumor, $\beta_2$ is the fixed slope for the predictor variable $X_2$ which is a vector of the purity of each tumor, $\beta_3$ is a change in the intercept for $X_3$ which is a categorical variable of individual genes within each proteostasis complex and $e$ is a Gaussian error term.

Unlike TCGA, samples within each cancer type in CCLE can be small and is unbalanced (i.e. some cancer types have <10 samples and others have >100 samples). In these cases, mixed effects models may not be able to estimate among-population variance accurately (*Harrison et al., 2018*). Thus, for all regression-based analyses in CCLE, a simple GLM was used instead. Cell viability values across all cell lines were z-score normalized by gene in all analyses to ensure fair comparisons across genes. To assess whether the same sets of genes are up-regulated in response to mutational load in CCLE using the GLM, a similar procedure to the GLMM was performed. A separate GLM was applied for each gene with the following parameters,

$$Y \sim \beta_0 + \beta_1 X_1 + e$$

where $Y$ is a vector normalized expression values of each cell line, $\beta_0$ is the fixed intercept, $\beta_1$ is the fixed slope for the predictor variable $X_1$ which is a vector of mutational load values for each tumor, and $e$ is a Gaussian error term. To assess whether protein abundances are similarly up-regulated in response to mutational load in CCLE in proteostasis complexes, a separate GLM was applied to each gene with the following parameters,

$$Y \sim \beta_0 + \beta_1 X_1 + \beta_2 X_2 + e$$

where $Y$ is a vector of protein abundance values within each cell line, $\beta_0$ is the fixed intercept, $\beta_1$ is the fixed slope for the predictor variable $X_1$ which is a vector of mutational load values for each tumor, and $e$ is a Gaussian error term. A similar GLM framework as above was used to estimate the association of mutational load and cell viability after the shRNA knock-down of individual genes in proteostasis complexes with the following parameters,

$$Y \sim \beta_0 + \beta_1 X_1 + \beta_2 X_2 + e$$

where $Y$ is a vector of normalized cell viability estimates after expression knock-down of an individual gene across all cell lines, $\beta_0$ is the fixed reference intercept, $\beta_1$ is the fixed slope for the predictor variable $X_1$ which is a vector of mutational load values for each cell line, $\beta_2$ is a change in the intercept for $X_2$ which is a categorical variable of individual genes within each proteostasis complex, and $e$ is a Gaussian error term. To estimate the association of mutational load and cell viability after pharmacologic inhibition of proteostasis machinery, the following GLM was applied to each relevant drug in PRISM:

$$Y \sim \beta_0 + \beta_1 X_1 + e$$

where $Y$ is a vector normalized cell viability estimates after drug inhibition across all cell lines, $\beta_0$ is the fixed intercept, $\beta_1$ is the fixed slope for the predictor variable $X_1$ which is a vector of mutational load values for each tumor, and $e$ is a Gaussian error term.

## Model validation

For both the GLM and GLMM, model assumptions of homogeneity of variance were verified by plotting residuals versus fitted values in the model and residuals versus each covariate in the model. Multi-collinearity with other mutational classes (e.g. such as CNAs) were considered but not found to correlate with point mutations (*Figure 1—figure supplement 1*). A jackknife re-sampling procedure was used for outlier analysis and to determine whether specific cancer types are driving patterns of association within the GLM and GLMM. Briefly, each cancer type was removed from the regression model one at a time, and regression coefficients were re-estimated. Overall, regression coefficients were fairly stable across cancer types and trends remained the same (*Figure 3—figure supplements 1 and 4*).

## Proteostasis gene sets

Genes for chaperone complexes were identified from *Hadizadeh Esfahani et al., 2018* and genes that are co-chaperones were not considered. Proteasome and ribosomal complexes were identified from CORUM (*Giurgiu et al., 2019*).

## Gene set enrichment analysis

All gene set enrichment analysis was performed using gprofiler2 with default parameters. For all sets of genes, significance was determined after correcting for multiple hypothesis testing (FDR<0.05). For gene set enrichment analysis used to identify genes up-regulated in TCGA in response to mutational load, all terms in the CORUM database were reported and enrichment terms in the KEGG database of diseases not related to cancer (e.g. 'Influenza A') were omitted from the main figures for clarity and space. For gene sets used to identify terms differentially spliced in between high and low mutational load tumors, all terms in the CORUM and the REACTOME database were reported in the main figures. The full set of enrichment terms for all analyses is reported in *Supplementary file 1*.

## Alternative splicing analysis

Alternative splicing events were quantified through a previously established metric called PSI. PSI is calculated as the number of reads that overlap the alternative splicing event (e.g. for intron retention, either at intronic regions or those at the boundary of exon to intron junctions) divided by the total number of reads that support and don't support the alternative splicing event. PSI summarizes alternative splicing events at specific exonic boundaries in the entire transcript population without needing to know the complete underlying composition of each full-length-transcript.

Alternative splicing calls for all tumors in TCGA were downloaded from TCGA SpliceSeq (*Ryan et al., 2016*). Default splice event filters (percentage of samples with PSI values >75%) from the database were applied. To test whether gene expression is down-regulated in high mutational load tumors through alternative splicing, we calculated whether alternative splicing events were present based on different threshold values of percent spliced in (PSI) from 90–50% (*Figure 2—figure supplement 2*). For each alternative splicing event in a gene, we quantified whether the expression of the same gene was under-expressed. Each gene was counted as under-expressed if it was one standard deviation below the mean expression within each cancer type. Genes that contained a point mutation within the same alternative splicing event were removed to control for eQTL effects. We note that intron retention events removed from this analysis represent only ~1% of intron retention events across all tumors and similar trends are found when this filtering scheme is not applied (*Figure 2—figure supplement 1*). In addition, we evaluated whether this trend is robust to other alternative splicing events (i.e. Alternate Donor Sites, Alternate Promoters, Alternate Terminators, Exon Skipping Events, ME = Mutually Exclusive Exon; *Figure 2—figure supplement 2*).

To investigate which genes are differentially spliced in between low and high mutational load tumors for specific alternative splicing events (i.e. intron retention), a t-test was used to calculate whether PSI values were significantly different in tumors with <10 protein-coding mutations compared to tumors with >1000 protein-coding mutations. Each alternative splicing event within a gene was required to have less than 25% of missing PSI values and a mean difference between the two groups of >0.01 to be considered. This approach identified 606 and 201 significant genes that have more and fewer intron retention events in high mutational load tumors, respectively, after correcting for multiple hypothesis testing (FDR<0.05).

## Drug category annotation and enrichment analysis

A separate GLM was ran for all drugs in the PRISM database to evaluate whether they are associated with mutational load and cell viability. All drugs that were negatively associated with mutational load and viability were queried on PubMed based on their reported mechanism of action in PRISM and grouped into broad categories (*Supplementary file 1*). Categories of drug mechanism of action were first chosen based on their role in metabolism and known hallmarks of cancer. Additional categories not directly related to known cancer-associated functional groups were made for drugs that could not otherwise be grouped (i.e. 'Ion Channel Regulation,' Viral Replication Inhibitor,' etc.). Drugs with an ambiguous mechanism of action (e.g. 'cosmetic,' 'coloring agent') were grouped into 'Other.' The abstracts of up to 10 associated papers were used to examine for evidence connecting drug

mechanisms of action to 33 broad categories. In total, 700 drug mechanisms of action were grouped and annotated into 33 broad categories. These broad categories were used to assess whether high mutational load cancer cell lines are generically vulnerable to drugs or whether certain categories are more likely to contain drugs effective against high mutational load cell lines. To control for differences in the number of drugs within each category, 50 drugs were randomly sampled, and the fraction of drugs significantly associated with mutational load in each category was calculated 100 times to generate confidence intervals.

## Acknowledgements

We thank Kathleen Houlahan, Chuan Li, José Aguilar-Rodríguez, and other members of Petrov and Curtis labs for their helpful comments and discussions. ST was supported in part by an NIH training grant T32-HG000044-21. This work was supported in part by the National Institutes of Health (NIH) Director's Pioneer Award: DP1CA238296 and the National Cancer Institute (NCI) Cancer Target Discovery and Development Center (U01CA217851) to CC and by NIH grants R35GM118165 to DAP and GM74074 to JF.

## Additional information

### Competing interests

Christina Curtis: Advisor and stockholder in GRAIL, Ravel, DeepCell and advisor to Genentech, Bristol Myers Squibb, 3T Biosciences and Nanostring. Dmitri A Petrov: Founder of, and stockholder equity in, D2G Oncology Inc. The other authors declare that no competing interests exist.

### Funding

| Funder | Grant reference number | Author |
|---|---|---|
| National Institutes of Health | GM74074 | Judith Frydman |
| National Institutes of Health | R35GM118165 | Dmitri A Petrov |
| National Institutes of Health | DP1CA238296 | Christina Curtis |
| National Cancer Institute | U01CA217851 | Christina Curtis |
| National Institutes of Health | T32-HG000044-21 | Susanne Tilk |

The funders had no role in study design, data collection and interpretation, or the decision to submit the work for publication.

### Author contributions

Susanne Tilk, Conceptualization, Software, Formal analysis, Visualization, Writing – original draft, Writing – review and editing; Judith Frydman, Conceptualization; Christina Curtis, Conceptualization, Supervision, Writing – review and editing; Dmitri A Petrov, Conceptualization, Supervision, Writing – original draft, Writing – review and editing

### Author ORCIDs

Susanne Tilk ![ORCID] https://orcid.org/0000-0002-9156-9360
Judith Frydman ![ORCID] https://orcid.org/0000-0003-2302-6943
Christina Curtis ![ORCID] https://orcid.org/0000-0003-0166-3802
Dmitri A Petrov ![ORCID] https://orcid.org/0000-0002-3664-9130

Joint Public Review: https://doi.org/10.7554/eLife.87301.2.sa1
Author response https://doi.org/10.7554/eLife.87301.2.sa2

## Additional files

### Supplementary files
• Supplementary file 1. The full set of enrichment terms for all analyses.

• MDAR checklist

### Data availability
Whole-exome, somatic mutation calls of 10,486 cancer patients across 33 cancer types in The Cancer Genome Atlas (TCGA) were downloaded from the Multi-Center Mutation Calling in Multiple Cancers (MC3) project (*Ellrott et al., 2018*) (https://gdc.cancer.gov/about-data/publications/mc3-2017). For the same patients in TCGA, RNA-seq data of log2 transformed RSEM normalized counts were downloaded from the UCSC Xena Browser (*Goldman et al., 2020*) (https://xenabrowser.net/datapages/) and copy number alterations (CNAs), including amplifications and deletions, called via ABSOLUTE were downloaded from COSMIC (v91) (*Bamford et al., 2004*) (https://cancer.sanger.ac.uk/cosmic/download). Tumor purity estimates for TCGA were downloaded from the Genomic Database Commons (GDC) (*Grossman et al., 2016*) (https://gdc.cancer.gov/about-data/publications/pancanatlas). Data for all cancer cell lines in the Cancer Cell Line Encyclopedia (CCLE) were downloaded from DepMap (*Barretina et al., 2012*) (https://depmap.org/portal/download/all/). Specifically, mutation calls (Version 21Q3) from whole-exome sequencing data, copy number alternations quantified by ABSOLUTE (Version CCLE 2019), log2 transformed TPM normalized counts (Version 21Q3) from RNA-seq data, proteomics data quantified by mass spectrometry (*Nusinow et al., 2020*), shRNA data from project Achilles (*Tsherniak et al., 2017*) normalized using DEMETER (DEMETER2 Data v6), and primary drug sensitivity screens of replicate collapsed log fold changes relative to DMSO from project PRISM (*Corsello et al., 2020*) (Version 19Q4) were used. All code used for analysis is publicly available on GitHub under the open-source MIT at https://github.com/stilk/protein, copy archived at *Tilk, 2022*.

The following previously published datasets were used:

| Author(s) | Year | Dataset title | Dataset URL | Database and Identifier |
|---|---|---|---|---|
| TCGA, PanCanAtlas | 2017 | Scalable open science approach for mutation calling of tumor exomes using multiple genomic pipelines | https://gdc.cancer.gov/about-data/publications/mc3-2017 | NCBI Genomic Data Commons, mc3-2017 |
| Tate JG, Bamford S, Jubb HC, Sondka Z, Beare DM, Bindal N, Boutselakis H, Cole CG, Creatore C, Dawson E, Fish P, Harsha B, Hathaway C, Jupe SC, Kok CY, Noble K, Ponting L, Ramshaw CC, Rye CE, Speedy HE, Speedy HE, Stefancsik R, Thompson SL, Wang S, Ward S, Campbell PJ, Forbes SA | 2018 | COSMIC | https://cancer.sanger.ac.uk/cosmic/download | Catalogue of Somatic Mutations in Cancer, v91 |

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
