## [Editor Report · eLife assessment]

Tilk and colleagues present a computational analysis of tumor transcriptomes to investigate the hypothesis that the large number of somatic mutations in some tumors is detrimental such that these detrimental effects are mitigated by an up-regulation by pathways and mechanisms that prevent protein misfolding. The authors address this question by fitting a model that explains the log expression of a gene as a linear function of the log number of mutations in the tumor and show that specific categories of genes (proteasome, chaperones, ...) tend to be upregulated in tumors with a large number of somatic mutations. Some of the associations presented could arise through confounding, but overall the authors present **solid** evidence that mutational load is associated with higher expression of genes involved in mitigation of protein misfolding – an **important** finding with general implications for our understanding of cancer evolution.

---

## [Referee Report · Joint Public Review]

Tilk and colleagues present a computational investigation of tumor transcriptomes to investigate the hypothesis that the large number of somatic mutations in some tumors is detrimental and that these detrimental effects are mitigated by an up-regulation by pathways and mechanisms that prevent protein misfolding.

The authors address this question by fitting a model that explains the log expression of a gene as a linear function of the log number of mutations in the tumor and additional effects for tumor homogeneity and type. This analysis identified a large number of genes (5000) that are more highly expressed at high mutational load at a FDR of 0.05. These genes are enriched in many core categories, most prominently in the proteasome, translation, and mitochondral translation. The authors then proceed to investigate specific categories of upregulated genes further.

The individual reviews, and the discussion among the reviewers, raised several issues that could potentially undermine or weaken some of the findings presented in this paper.

1. Systematic differences in expression of some genes from one tumor class to another might generate spurious associations with mutational load (ML), which would affect the results presented in Figs 1 and 3. The case of a causal link between ML and over-expression of genes that mitigate deleterious effects of misfolding would be stronger if these results were replicated within single cancer types with many samples with different ML (similar to how Fig S6 relates to Fig 3). A related concern might be an association between increased variance of expression and ML. The compositional nature of expression data could generate trends like the ones shown in Fig. 2 with changing variance.

2. Fig 4, Fig S5 and Fig S8 show results for the regression coefficient of expression on ML after leaving out one cancer at a time. All of us initially read this as results for 'one cancer at a time', rather than 'leave-one-out'. These figures are used to argue that the results are not driven by specific cancer types. However, this analysis would not reveal if the signal was driven by a (small) subset of cancer types. To justify claims like "significant negative relationship between mutational load and cell viability across almost all cancer types", one needs to analyze individual cancer types. Results for specific genes, rather than broad groups would also help interpret these results.

3. You use different model architecture for the TCGA and CCLE analysis because you suspect that the sample size imbalance in the latter might mean that a GLMM can not capture the different variance components accurately. Did you test this? Could you downsample to avoid this? Cancer type is likely a strong confounder of ML.

4. In the splicing analysis (Fig 2 and Fig S4), you report a 10% variation in splicing for a 100-fold variation in ML. This weak trend is replicated in very similar ways for many different types of alternative splicing events. It is not clear why different events (exon skipping, intron retention, etc) should respond in the same way to ML. A weak but homogeneous effect like the one shown here might result from some common confounder (see point 1). Similarly, it is not clear why with increasing intron retention PSI threshold the fraction of under-expressed transcripts would decrease and not increase.

---

## [Author Response]

eLife assessmentTilk and colleagues present a computational analysis of tumor transcriptomes to investigate the hypothesis that the large number of somatic mutations in some tumors is detrimental such that these detrimental effects are mitigated by an up-regulation by pathways and mechanisms that prevent protein misfolding. The authors address this question by fitting a model that explains the log expression of a gene as a linear function of the log number of mutations in the tumor and show that specific categories of genes (proteasome, chaperones, ...) tend to be upregulated in tumors with a large number of somatic mutations. Some of the associations presented could arise through confounding, but overall the authors present solid evidence that mutational load is associated with higher expression of genes involved in mitigation of protein misfolding – an important finding with general implications for our understanding of cancer evolution.

We thank the reviewers for these kind words. The summary statement and public review highlight our work in understanding how human tumors phenotypically respond to mutational load by assessing changes in gene expression. This work provides a mechanistic underpinning to our previous finding that the accumulation of passenger mutations in tumors creates a substantial cost because even substantially damaging passenger mutations can fix in non-recombining clonal tumor lineages. At the same time, we believe the summary statement and the public review do not mention a key remaining part of our paper that validates our findings and establishes causal connections between protein misfolding due to coding passenger mutations and tumor fitness. Specifically, we replicate and cross-validate our findings in human tumors by examining expression responses in an independent dataset of cancer cell lines (CCLE), where we demonstrate similar expression responses to an accumulation of mutations, indicating generic, cell intrinsic responses. We then establish a causal link by demonstrating that mitigation of protein misfolding through protein degradation and re-folding is necessary for high mutational load cancer cells to maintain viability through perturbation experiments via shRNA known-down and treatment with targeted agents. These analyses and results are important because they show that the adaptive responses we observe are evidence of a generic, cell intrinsic phenomenon that cannot be explained by organismal effects, such as aging, changes in the immune system or microenvironment.

Joint Public Review:Tilk and colleagues present a computational investigation of tumor transcriptomes to investigate the hypothesis that the large number of somatic mutations in some tumors is detrimental and that these detrimental effects are mitigated by an up-regulation by pathways and mechanisms that prevent protein misfolding.The authors address this question by fitting a model that explains the log expression of a gene as a linear function of the log number of mutations in the tumor and additional effects for tumor homogeneity and type. This analysis identified a large number of genes (5000) that are more highly expressed at high mutational load at a FDR of 0.05. These genes are enriched in many core categories, most prominently in the proteasome, translation, and mitochondral translation. The authors then proceed to investigate specific categories of upregulated genes further.The individual reviews, and the discussion among the reviewers, raised several issues that could potentially undermine or weaken some of the findings presented in this paper.1. Systematic differences in expression of some genes from one tumor class to another might generate spurious associations with mutational load (ML), which would affect the results presented in Figs 1 and 3. The case of a causal link between ML and over-expression of genes that mitigate deleterious effects of misfolding would be stronger if these results were replicated within single cancer types with many samples with different ML (similar to how Fig S6 relates to Fig 3). A related concern might be an association between increased variance of expression and ML. The compositional nature of expression data could generate trends like the ones shown in Fig. 2 with changing variance.

We agree with the reviewers that possible confounders should be considered since TCGA data is heterogeneous. In this paper, we investigated possible confounders such as multicollinearity with different mutational types (SNVs and CNVs), controlled for expression responses within cancer types in the GLMM, and used the jackknifing procedure to ensure that no one cancer type dominates the signal. However, in principle unknown hidden confounders could remain, which is why a large part of our paper was focused on validating these effects in an independent dataset (CCLE) where many other covariates are not relevant (immune system, donor variability, stage, age, sex, etc.). Importantly, we also used data from perturbation screens that are completely orthogonal to expression responses in CCLE to get at a cause and effect.

Our reasoning for using all of the data in Figure 1 while controlling for differences due to cancer type in the GLMM was to maximize the variation in mutational load across all of the samples in this dataset to identify what genes increase in expression as mutational load increases over 5 orders of magnitude. As noted here, we also already further validated that the signal we observe in Figure 1 is still robust for our gene sets of interest within cancer types in Supplemental Figure 6.

1. Fig 4, Fig S5 and Fig S8 show results for the regression coefficient of expression on ML after leaving out one cancer at a time. All of us initially read this as results for 'one cancer at a time', rather than 'leave-one-out'. These figures are used to argue that the results are not driven by specific cancer types. However, this analysis would not reveal if the signal was driven by a (small) subset of cancer types. To justify claims like "significant negative relationship between mutational load and cell viability across almost all cancer types", one needs to analyze individual cancer types. Results for specific genes, rather than broad groups would also help interpret these results.

Our reasoning for grouping together genes in Figure 4 was because the shRNA screen was done on a single gene at a time, and we were interested in measuring the joint effect on viability after knocking down all of the genes in a given complex.

Given that the expression responses in Figure 3 already validate within cancer types in TCGA in Supplemental Figure 6, we believe that it’s very unlikely that the signal we observe is driven by individual cancer types or smaller groups of cancer types. In addition, we did not perform a within cancer analysis in CCLE for Figure 4, because not all available cancer types in CCLE were profiled evenly in the shRNA screen (Total < 300). The vast majority of cancer types in CCLE for the shRNA screen (23/26) have sample sizes <20 within each group that we believe are unlikely to lead to meaningful results that are not driven by noise.

1. You use different model architecture for the TCGA and CCLE analysis because you suspect that the sample size imbalance in the latter might mean that a GLMM can not capture the different variance components accurately. Did you test this? Could you downsample to avoid this? Cancer type is likely a strong confounder of ML.

That was indeed our reasoning, that within group sample sizes in CCLE are too low to robustly estimate variance within cancer types. Given that many cancer types have <20 samples within each group, we don’t think that evenly downsampling would enable us to get an estimate not driven by noise. As noted above, our approach to control for this was to perform a jackknifing procedure that eliminates a single cancer type at a time and re-estimates the effect.

1. In the splicing analysis (Fig 2 and Fig S4), you report a 10% variation in splicing for a 100-fold variation in ML. This weak trend is replicated in very similar ways for many different types of alternative splicing events. It is not clear why different events (exon skipping, intron retention, etc) should respond in the same way to ML. A weak but homogeneous effect like the one shown here might result from some common confounder (see point 1). Similarly, it is not clear why with increasing intron retention PSI threshold the fraction of under-expressed transcripts would decrease and not increase.

We agree that the effects of all the different alternative splicing effects are complex. Our focus was on intron retention, which is known to occur in cancer (Lindeboom, et. al 2016, Nature Genetics), and our analysis is consistent with the idea that damaging passenger mutations can shift cellular phenotypic states that require the use of many different mechanisms to mitigate protein misfolding.

For Figure S4, as the PSI threshold for calling an alternative splicing event increases, fewer samples are called as having an intron retention event in the gene. This uniformly decreases the numerator across all the mutational load bins, so that when the threshold is increased the fraction of under-expressed transcripts with intron retention events is lower.